# Genomic Fishing and Data Processing for Molecular Evolution Research

**DOI:** 10.3390/mps5020026

**Published:** 2022-03-07

**Authors:** Héctor Lorente-Martínez, Ainhoa Agorreta, Diego San Mauro

**Affiliations:** Department of Biodiversity, Ecology and Evolution, Complutense University of Madrid, 28040 Madrid, Spain

**Keywords:** genomics, high-throughput sequencing, data mining, gene family, blast search, sequence alignment, phylogeny, molecular evolution

## Abstract

Molecular evolution analyses, such as detection of adaptive/purifying selection or ancestral protein reconstruction, typically require three inputs for a target gene (or gene family) in a particular group of organisms: sequence alignment, model of evolution, and phylogenetic tree. While modern advances in high-throughput sequencing techniques have led to rapid accumulation of genomic-scale data in public repositories and databases, mining such vast amount of information often remains a challenging enterprise. Here, we describe a comprehensive, versatile workflow aimed at the preparation of genome-extracted datasets readily available for molecular evolution research. The workflow involves: (1) fishing (searching and capturing) specific gene sequences of interest from taxonomically diverse genomic data available in databases at variable levels of annotation, (2) processing and depuration of retrieved sequences, (3) production of a multiple sequence alignment, (4) selection of best-fit model of evolution, and (5) solid reconstruction of a phylogenetic tree.

## 1. Introduction

The relatively low cost and increasing power of modern (high-throughput) sequencing technologies have resulted in a massive increase of the number of genome projects on non-model organisms [1,2]. This has prompted the development of numerous databases to store all the generated data. As a logical consequence, figuring out how to extract relevant information from such vast amounts of data becomes fundamental. In this context, bioinformatics has emerged as a key tool for handling and processing sequence data derived from genome-scale sequencing experiments. Proteomics, genomics, transcriptomics, and other new disciplines have emerged as a result of the fusion of programming languages and cutting-edge sequencing technologies. However, gene sequence retrieval and assessment are still arduous and complex processes. High-throughput sequencing (HTS) platforms based on DNA amplification, such as Illumina, typically yield short reads of around 100 base pairs [2,3]; hence, systematic assembly of the data (e.g., reads to contigs, contigs to scaffolds, etc.) is mandatory to obtain final sequences of genes or genomic regions. Molecule-sequencing platforms, such as PacBio or NanoPore, can yield much longer reads, thus reducing the gap between sequencing output data and real gene or genomic sequences [2,4]. In all cases, the identification and annotation of relevant and meaningful genomic regions always remains a mandatory step, and the comparative analysis of genome sequences is central to such an endeavour [5]. Basically, conserved functions between two organisms are assumed to be encoded in DNA in a similar way. Therefore, similar DNA, RNA, or protein sequences are likely involved in relatively similar functions and assumed to be homologous (orthologues or paralogues). In this context, comparative genomics can make use of sequence alignment and phylogenetic analysis as a framework to try to understand the evolutionary processes that trigger sequence diversification. Knowing the pattern of historical relationships among groups (lineages) of elements (organisms, sequences) allows for possible biases and dependencies derived from shared ancestry to be amended when interpreting a function, structure, or any other pattern involving genes or genomic regions [6].

Nevertheless, the pathway from raw reads to gene alignments and phylogenetic trees is not necessarily straightforward but rather challenging and often very intense in terms of both time and resources (e.g., computing power). In the last few years, a number of programs and pipelines for relatively automated extraction of relevant information from modern sequencing technology outputs have been developed. Of the several tools available, software such as geneid [7], Prokka [8], or GenMark [9] allow for complete gene mapping all across the genome. In a similar way, several platforms have appeared for RNA-seq analysis, such as the TRUFA web server [10], eventually intended for sequence assembly (either de novo or referenced) and gene annotation and alignment. In terms of gene isolation, BLAST [11] is certainly the most broadly used tool, but other programs, such as ORTHOSCOPE [12], make use of it for identification and isolation of groups of related orthologs. In general, all these pipelines and web servers are intended for analysis of genomic and/or HTS data oriented toward gene mapping and identification; however, to our knowledge, direct processing and preparation of resulting outputs for molecular evolution analysis are lacking.

In this study, we provide a protocol and pipeline for gene search and capture/isolation (fishing) for particular sets of organisms (at any degree of taxonomic diversity) from large-scale genomic data retrieved from publicly available databases. The isolated gene sequences are subsequently aligned and submitted to robust phylogenetic reconstruction using adequate modelling of the substitution process. Altogether, this constitutes baseline data for conducting molecular evolution analyses, such as detection of adaptive/purifying selection or ancestral protein reconstruction. The protocol described here is relatively unique in its span: from genomic mining to phylogeny reconstruction in a comprehensive step-by-step workflow.

Our protocol is intended to be a cross-platform workflow that can be executed on Linux, macOS, and Windows machines. As mentioned above, programming languages are very useful (often mandatory) when working with genomic-scale data. Therefore, working with command-line interfaces on system terminals (such as those of Linux- and Unix-based macOS) becomes a sensible choice in practice. In order to facilitate particular steps of the process, we have developed a suite of small Python programs (GNFish package), taking advantage of the free-access Biopython project environment [13]. We strongly recommend using this package, especially for programming language beginners. As an alternative itinerary for the protocol (e.g., for those preferring not to deal with command-line procedures), we additionally describe most steps using windows-style interfaces with local programs or remote/online web tools. The protocol presented here is perfectly adequate for research studies on just one or multiple gene families alike. Theoretically, it can also be used for complete gene mapping all across the genome, although there are more specific tools and pipelines publicly available for this purpose (such as Prokka [8] or geneid [7], as detailed above). A schematic flowchart of the main steps of the protocol is shown in Figure 1.

## 2. Experimental Design

### 2.1. Data Mining and Sequence Identification

Working with whole genome sequences is still an arduous task that is generally not affordable for users not familiar with programming language. A single genome can represent more than one gigabyte of data. Automated annotation programs have facilitated the identification and extraction of protein-coding sequences by mapping potential open reading frames in the genomes to place gene locations. However, the annotation process is not evenly spread among all the available genome data. In some cases, only preliminary rounds of assembly have been developed, resulting in records that contain multiple scaffold entries. These scaffolds correspond to huge portions of the genome sequence in which gene identification has not been fully accomplished yet. Even coding RNA or protein datasets derived from genomic data can contain thousands of entries. Therefore, establishing a solid protocol for specific gene identification is essential when working with genomic data. This becomes of particular interest for research oriented toward molecular evolution insights at any taxonomic level. The first step of such a protocol consists of retrieving the relevant genomic data. This genomic data can be determined anew, although there is certainly a huge amount of genomic data already available and stored in publicly available databases and repositories (such as EukProt [14] or Ensembl [15]) that remains underused or at least not fully exploited for its entire potential. The National Center for Biotechnology Information (NCBI) comprises a comprehensive collection of connected databases, including gene, protein, genomic, and transcriptomic data, among others [16]. Within all the diversity covered by the different databases of the NCBI project, RefSeq emerges as a reliable choice because it maintains and curates a publicly available database of annotated genomic records [17]. Besides, the NCBI Entrez system provides an easy and powerful means to retrieve data [18].

When genomes are stored without annotations, a basic, straightforward method for gene identification can be implemented based on similarity searches of DNA and protein sequences, such as those performed by the BLAST algorithm for local alignment [19]. Basically, this tool is able to find related sequences in a database. Each genome can be treated as an independent database, and a list of query sequences of interest can be used as template. Typically, this query dataset is made of protein sequences, although the target corresponds to nucleotide sequences. Due to the degeneration of the genetic code, some of the changes that occur at the nucleotide level have no impact on the protein primary structure. Therefore, it is more plausible that the protein primary structure and sequence remain more conserved, thus facilitating gene identification. The RefSeq database provides reliable information for protein sequences, and the BLAST tool allows for a certain level of custom parametrization [11]. Of these, we simply explore/manipulate two: (1) *e-value*, a parameter that describes the number of hits that one can “expect” to see by chance when searching a database of a particular size, and (2) *outfmt*, which sets the output format.

BLAST output files must be parsed to extract the relevant information, but the specific procedure can vary depending on the type of data. For example, for protein and coding RNA data, every BLAST hit generally corresponds to a unique sequence. Therefore, it is enough to just extract the target-sequence match to obtain the gene sequence of interest. Based on our personal experience, these sequences are usually well annotated. Nonetheless, some hits can contain mismatched positions at the beginning or at the end of the sequences, especially when working with coding RNA sequences because they can present untranslated regions (UTR). On the other hand, working with whole genome sequences or big genomic portions (such as scaffolds) often implies a considerable challenge because of the huge amount of data to be handled and mined, as well as their intrinsically higher complexity. Indeed, single scaffolds typically comprise more than one gene; hence, BLAST output files must be thoroughly inspected to identify possible multiple-gene matches present in each file. Moreover, genomic-level sequences typically include the intronic fractions that must be removed to isolate coding and, ultimately, protein sequences. In this sense, a relatively simple an effective method to identify coding regions in genomic portions is to align these against some of the query sequences of interest as a template. There is a plethora of programs and algorithms to compute multiple-sequence alignment [20], but among them, MAFFT [21,22] stands among the most reliable and widely used, outperforming more classical programs, such as CLUSTALW or T-COFFEE [23,24]. Among its strengths, MAFFT implements several distinct algorithms, each adequate for different types of data and situations [21]. One of these, the E-INS-i algorithm, performs well for aligning coding-gene sequences that present intronic regions against closely related template sequences, such as those resulting from BLAST searches. Once the alignment is completed, the next step consists of cleaning up the (presumed) intronic portions (those mismatched by very long gaps) from the alignment to just isolate the coding-sequence regions. This trimming step can be conducted with general packages for sequence editing and manipulation, such as MEGA, which is a desktop application designed for comparative analysis of homologous gene sequences that allows for alignment visualization and modification with a user-friendly interface [25]. It may be the case that the process of alignment and trimming needs to be iteratively repeated and refined (progressively cleaning up intronic portions) until coding regions can be conclusively identified and isolated.

### 2.2. Phylogenetic Analysis

Studying the molecular evolution of genes (commonly across different organisms) is important to understand the biological processes in which they are involved. For this aim, first establishing a robust framework for the phylogenetic relationships of the different gene sequences involved provides the necessary groundwork that precedes further evolutionary analysis. In our particular context, the phylogenetic experimental design begins with the gathering of all gene sequences of interest (typically from a particular organism or group of organisms) into individual gene matrices (one for each gene of interest). At this point, it may be relevant to also include some external sequences (outgroups) if possible/available. These will provide information about ancestral states, serving as relative indicators of the direction of evolutionary change (e.g., which nodes are the oldest in the phylogenetic tree) [6]. The next step consists of aligning the sequences included in each of the matrices, which is crucial because all subsequent phylogenetic inferences rely on these alignments [26,27,28]. As mentioned above, MAFFT is our recommended choice for this task, either running an automatic strategy (which selects the most adequate alignment algorithm among those available depending on data size) or selecting a particular alignment algorithm (in this case, the L-INS-i may be a good option for sequences from the same gene family). Once the alignment is done, trimming ambiguously aligned positions can increase quality and, consequently, the reliability and accuracy of subsequent analyses [29]. trimAl is a tool for automated alignment trimming that is especially suited for large-scale phylogenetic analyses [30]. It is free and portable to all platforms, and it can be used online through the Phylemon web server [31]. trimAl implements modes for automated selection of trimming parameters, although the use of some can be computationally demanding, especially when working with very large datasets (a simpler option for genomic-scale data is the *conservation threshold* parameter based on the percentage of gaps).

Apart from robust alignments, proper characterization of the process of sequence evolution is essential in molecular phylogenetic inference [32] because phylogenetic methods tend to be less accurate or inconsistent when an incorrect model of sequence evolution is assumed [33,34]. Phylogenetic inference in a probabilistic framework, such as maximum likelihood (ML), allows for the estimation of complex substitution model parameters, branch lengths, and tree topology using heuristic methods. In this sense, the IQ-TREE software presents a set of fast and effective stochastic algorithms for ML phylogenetic analysis, including automated assessment of the best-fit substitution model [35]. IQ-TREE implements modern measures of branch support, such as the ultrafast bootstrap approximation approach (UFBoot) [36], which can reduce computing times compared to traditional bootstrap. Phylogenetic inference can be conducted at the nucleotide (DNA) or amino acid (protein) sequence level. In general, protein alignments (often obtained by conceptual translation of primary DNA records) are more adequate for inference of old relationships because the higher character-state space (20) of amino acids (compared to 4 of nucleotides) makes it less likely to observe homoplasy events due to sequence saturation [6]. The process of nucleotide or amino acid substitution is further complicated by the fact that the evolution of sites is often highly heterogeneous, with some sites changing rapidly, whereas others are highly conserved. In general, this heterogeneity of evolutionary rates among sites is modelled using specific parameters, such as a proportion of invariant sites or a discrete approximation (usually with four categories) of the continuous gamma function [37]. Furthermore, the substitution process can be affected by other factors, such as solvent exposure and secondary structure [38,39,40]; therefore, more complex models can be devised to better explain protein evolution [41]. Nevertheless, further discussion of these issues is outside the scope of the present study.

Although we developed our protocol with a preferred list of software for each of the different tasks of the process (retrieval of genomic data, gene sequence identification and isolation, multiple alignment, model selection, and phylogenetic inference), there are often alternatives that can do and perform equally well. We next discuss some of these alternatives. In the case of genome retrieval, EukProt is a database of published and publicly available predicted protein sets and unannotated genomes selected to represent eukaryotic diversity, including species from all major supergroups, as well as orphan taxa [14]. On the other hand, UniProt is a reference database for protein data that can be used to obtain the query sequences necessary for similarity searches [42]. For this purpose, HMMER can be used as an alternative to BLAST or even in combination with it. HMMER is based on probabilistic models called profile hidden Markov models (profile HMMs) [43], and it often works with protein profiles downloaded from Pfam [44] and Interpro [45] databases. Like BLAST, HMMER can also work with query sequences. In the case of multiple sequence alignments, CLUSTALW [23], T-COFFEE [24], and PRANK [46] are good alternatives to MAFFT. For phylogenetic inference, IQ-TREE is comparable to other ML programs, such as PhyML [47] and RAxML [48]. However, in these best-fit substitution models, selection is not automated (only in the online version of PhyML), and an external program is required for this task, such as ProtTest for protein alignments [49] or jModelTest for nucleotide alignments [50]. It is often the case that applying different best-fit models for distinct alignment sections that differ in rates of evolution (e.g., different genes, codon positions, stems vs. loops) might be preferred over averaging a single model for the entire set. Programs such as PartitionFinder [51] allow for simultaneous selection of best-fit partitioning strategy and substitution models, and this information can be easily implemented in RAxML and IQ-TREE. The latter implements an option for automated selection of partitions and models (see http://www.iqtree.org/doc/Advanced-Tutorial#partitioned-analysis-for-multi-gene-alignments, accessed on 27 January 2022).The MEGA package can also be used for phylogenetic inference, as well as for alignment and selection of best-fit model of substitution [25] but without taking into account partitioning schemes. Finally, phylogenetic trees can be graphically inspected and enriched using several publicly available programs, with FigTree [52] being our preferred choice because of its versatility and ability to produce publication-ready figures. A good alternative for this task may be Dendroscope [53].

## 3. Procedure

### 3.1. Data Mining and Sequence Identification

#### 3.1.1. Retrieval of Source Data (Genomic, RNA, and Protein)

A.Locally Using Python

Go to https://github.com/hectorloma/GNFish (accessed on 27 January 2022) to obtain the GNFish package. The “README.md” file contains a detailed explanation for the suite of Python programs used throughout this protocol, as well as some running examples.The following protocol details how to run Python programs using the Anaconda platform (See below). However, on the “README.md” file, you will find details on how to run it directly on a Linux terminal. Functionality and output files are the same.Download the main directory containing the scripts and examples by clicking on *Code → Download Zip* and decompress the file (Figure 2).Install Anaconda following the instructions at (https://conda.io/projects/conda/en/latest/user-guide/install/index.html, accessed on 27 January 2022) and open the Spyder program.Install Biophyton (if not installed yet). In the Spyder console (Figure 3), type *pip install biopython.*Again, in the Spyder console, type *cd* and drag *GNFish/Code* directory (remove quotes if you are a Windows user). Note that this step is mandatory every time you close the Spyder program.Type *run get_genomes.py -h* in the Spyder console to display help information and read the “README.md” file for further information and some running examples. This applies to all the “.py” programs used throughout this pipeline.Create a file with all your queries (usually species or higher taxon names). You can also add specific field tags (e.g., organism, assembly level, etc.) and some filters (e.g., latest [filter] or unambiguous [filter]).More information about how to concatenate specific field tags is found at https://www.ncbi.nlm.nih.gov/books/NBK3837/-EntrezHelp.Entrez_Searching_Options, accessed on 27 January 2022.For information about filters, check out https://www.ncbi.nlm.nih.gov/assembly/help/, accessed on 27 January 2022.For both filters and field tags, you can obtain more information after conducting a manual search. We recommend first trying a simple query with one taxa as an example following Section 3.1.1 B.Run the program typing *run get_genomes.py [e-mail address] [path to query file] [data-type]* on the Spyder console. Add any optional arguments after these mandatory ones.We recommend that you to use the *--refine* argument in order to refine your search. By default, this will apply *Latest*, *Representative, Not Anomalous* but you can add your own settings by typing them, enclosed by quotes, after the argument.Use the proper arguments according to your search. By default, the program will download whole-genome data. If downloading protein or RNA data, and they are not available, the program will try with the whole genome version. Stop that feature using --*exclusive* argument.Downloaded sequences will be stored into *Genomic*, *RNA* and *Protein* directories located at *GNFish/Code/Data*. Information about the downloaded genomes will be stored at *Data/downloaded_genomes_log.tsv* as well.

B.Through NCBI Website

Go to https://www.ncbi.nlm.nih.gov/assembly/?term=, accessed on 27 January 2022.Type your query (usually species or higher taxon names) on the search box located at the top of the web page (Figure 4).By clicking the *Advance* button right below the search box (Figure 4), you can manually add field tags (e.g., organism, assembly level, etc.).Filters *Latest* and *Exclude anomalous* are applied by default (Figure 4).After your search, you would find a side bar on the left with all the available filters (Figure 4).In addition, you can find a text box named “Search details” on the right with the specific command of your query (Figure 4).This text box can be useful for creating custom queries that can be used in the automatic path (See Section 3.1.1 A).Note that when there are more than one field tag or filter, they appear enclosed in parentheses (Figure 4).We recommend keeping the default filters and adding *Representative or Reference* from *RefSeq category* by clicking on the left side panel (Figure 4) or by typing *AND representative genome [filter] OR reference genome [filter]* within the filter parentheses in the search box.Click on the *Download Assemblies* button to download the assembly (Figure 5).As mentioned above, genome assembly may not be evenly distributed. This can be a problem when downloading.We recommend that you to choose *RefSeq* under “Source database” (Figure 5) and either *Protein FASTA (.faa)* or *RNA FASTA (.fna)* under “File type”.After decompressing the downloaded file, you will find a directory tree similar to *genome_assemblies_genome_fasta/ncbi-genomes-date*. Within this, you will find all the files for every species.Then, repeat the search, but this time, when downloading, choose *GenBank* under “Source database” (Figure 5) and *Genomic FASTA (.faa)* under “File type”. Right after that, check what species were previously downloaded, and then check just the boxes of the remaining species.Of course, these steps are not intended for downloading a large number of genomes; for such purposes, we recommend following Section 3.1.1 A.If there is just one assembly for the selected taxa, you do not need to apply any filter. The web will automatically redirect you to the assembly entry page (Figure 6).We recommend storing every “.gz” file in a separate folder according to its data type (*Genomic*, *RNA*, *Protein*) first and then according to the species/organism.Download the file by clicking on *Download Assemblies* (Figure 6) as detailed above.

#### 3.1.2. Generation of Query Sequences

A.Locally Using Python

Type *run get_query_sequences.py [email address] [path to query file] --curated --refine*. This will download a protein dataset containing 200 sequences that includes the name in “Protein Feature” (curated parameter) from “RefSeq” (refine parameter).The “query.txt” file can include several queries. The programs expect a gene name, with fields and filters enclosed in parentheses right after it.You can type your own field tags and filters, typing them after *--refine* argument in a similar way as when downloading genomes (See Section 3.1.1).In addition, you can restrict the number of downloaded sequences to a maximum number using *--retmax* arguments. When using *--curated* arguments, the program should curate sequences based upon this number; therefore, you may obtain a smaller number of sequences than with *--retmax*.There is not a perfect number of query sequences. Ideally, the best number should maximize the diversity of the studied gene family and minimize computing time. Our approach (200) aims for a great coverage of this diversity.BLAST can perform well with just a few sequences (around 10), reducing computing time. Therefore, another strategy could be to manually select some key sequences and download them one at a time. Of course, this requires a solid knowledge of the studied gene family.All this is for protein downloading, recommend as query when using BLAST searches. However, download of nucleotide sequences is also allowed (if needed for alignment; see below). However, this is not refined, so we recommend using *biomol_mrna[PROP]* to download just the transcripts.The database will be stored at *Data/Query_seqs*, and its name will be the name of the gene you entered, followed by “query_sequences_data_type.fas”.

B.Through NCBI Website

Go to https://www.ncbi.nlm.nih.gov/protein/?term=, accessed on 27 January 2022, (Figure 7).Type the name of your protein in the search box at the top (Figure 7).In a similar way as explained for downloading genomes (see above), you can manually add field tags by clicking the *Advance* button right below the search box (Figure 7). You can also add filters to refine your search.We recommend that you use the default *refseq[filter]* (Figure 7). In the “Search box” on the right, you can look at the command that you are applying to your search (Figure 7).To download the database, click on *Send to* → *File* → *Format* → *FASTA* → *Create File* (Figure 7).Note that this will download the whole list of results. Therefore, it is important to be as precise as possible with your query.You can download nucleotide sequences in a similar way at https://www.ncbi.nlm.nih.gov/nuccocre/?term=, accessed on 27 January 2022. Type *biomol_mrna[PROP]* after your query to download transcripts.

#### 3.1.3. BLAST Searches

A.Locally Using Python

Type *run decompress_genomes.py* in the Spyder Python console to decompress the genomic data files. Use *genomic*, *RNA*, or *protein* arguments or *directory* for your own custom path.Go to https://ftp.ncbi.nlm.nih.gov/blast/executables/blast+/LATEST/ (accessed on 27 January 2022) and download ncbi-blast-version-x64-win64.tar.gz (Windows users) or ncbi-blast-version-x64-linux.tar.gz (Linux and Mac users).Decompress the BLAST program.Run *blast.py [blast_directory] [data_type].*BLAST programs are located in the *bin* folder inside the BLAST directory. Drag the *bin* folder to the Spyder console after *--blast_path* parameter when running.The program will check in *Genomic*, *RNA*, and *Protein* directories automatically. You can change the directory by using --*directory* arguments, but you must also specify the genomic data type.You can modify the e-value parameter (see https://www.ncbi.nlm.nih.gov/books/NBK279690/, accessed on 27 January 2022, and https://www.ncbi.nlm.nih.gov/books/NBK279684/table/appendices.T.options_common_to_all_blast/ for more information, accessed on 27 January 2022). You will obtain a “.tsv” file with all the hits found in your target genomic data.

#### 3.1.4. Extraction of RAW Sequences

A.Locally Using Python

Type *run get_unique_hits.py* to obtain the best hit for every of the entries of your target data.For whole-genome data, go to Section 3.1.4. Multiple Gene Inspection below and then continue with the next step.Type *run get_RAW_sequences.py [data_type]* to extract every sequence corresponding to each unique hit. The extracted sequences will be stored in the *Extraction* directory located in the same folder as the whole genome file.Change directory using --*directory* argument but keep using genomic data type.If using whole genome sequences, you may have to modify the *--in_len* parameter to control the intron length.Using *--query_seqs* arguments and typing your database file path allows you to attach some of the database sequences that match your query entry.By default, the program will attach the five (arbitrary number) first non-redundant sequences according to the entries of the BLAST output file. Change this using *--query_seqs_num* parameter.Note that for whole genome entries that include more than one gene, this number depends on the number of modified query entry IDs (see above).You can manually add any sequence. Preferably, add sequences from closely related species. You can download single sequences in the same way as the query dataset (see Section 3.1.2).

B.Locally Using a Text Editor

For whole-genome data go to Section 3.1.4. Multiple Gene Inspection below and then continue with the next step.Open every “whole_genome_name_out.tsv” file. Look at the second column (target ID) and keep just unique IDs. For whole-genome data, follow Section 3.1.4 A.Open your genomic data (i.e., *Genus_species_id.faa*) with a text editor. This step cannot be performed in some cases, especially those that imply the use of non-annotated genome sequences.Pick up every unique target ID and search for it in the corresponding genomic data file.

Multiple Gene Inspection (Mandatory for Whole-Genome Data; Skip if Using RNA or Protein Data).

The following steps assume that you have used *--outfmt* 6 (i.e., BLAST tabular output format 6).Open the “whole_genome_name_out.tsv” file with a spreadsheet program (such as Microsoft Excel) or a text editor.Every row corresponds to a different hit, and the second column indicates the target identifier (scaffold ID) (Figure 8).Controlling by the 2nd column, you must check the 9th and 10th columns that contain the start and end positions where the query sequence aligned within the target entry and compare entries below to identify different start or end positions that could be associated with two different genes.In order to facilitate visibility, you can highlight every target by clicking on *Conditional Formatting* → *Highlight Cells Rules* → *Equal to* → *Choose the corresponding cell* → *OK*.You can also highlight the 9th and 10th columns in the same way using *Between to* instead of *Equal to* for controlling.If you spot another gene in the same query entry, you must modify the query entry ID (first column) by adding “_1” to the rows that belong to the first one, “_2” to the second one, etc. We recommend changing at least five query entry IDs if possible in order to facilitate proper gene fishing (see below). Additionally, you must update the “_unique.tsv” file with the new query names, and you must add at least one row containing the information of the new gene(s).Before continuing with the next target ID, click on *Conditional Formatting* → *Clear Rules* → *Clear Rules for Entire set*.

#### 3.1.5. Coding Sequence Identification (Although This Step Is Mandatory When Working with Whole Genome Sequences, You Can Skip It When Working with Protein and RNA Sequences)

You should have used --query_seqs earlier (Section 3.1.4 A) to attach template sequences or have manually added some.

##### Alignment

A.Locally Using Python

Go to MAFFT software web (https://mafft.cbrc.jp/alignment/software/, accessed on 27 January 2022) and navigate to the specific page according to your operating system. Follow the steps to install MAFFT software on your computer.On the Spyder Python console, type *run align_sequences.py*. You can choose a specific alignment algorithm using *--algorithm*. The MAFFT manual recommends using the *Auto* function when you know little about your data. For genomic data and working with one gene family, we recommend using the *E-INS-i* algorithm.As in other cases, you can use *genomic*, *rna*, and *protein* or *directory* arguments.Windows users may encounter some problems in either installing or running MAFFT, especially those using older system versions. If this is the case, look at the next section.

B.Through MAFFT Server

Go to the MAFFT online version page (https://mafft.cbrc.jp/alignment/server/, accessed on 27 January 2022).Paste the content of the file you want to align in the available text box or browse for your file by clicking on the *Choose File* button.Select *Same as input* for the options: *UPPERCASE/lowercase*, *Direction of nucleotide sequences*, and *Output order.*Scroll down to the *Advance settings* section. In the *Strategy* section, we recommend using *Auto* if you know little about your data. For genomic data, we recommend using the *E-INS-i* algorithm.Download the alignment from the results page (Figure 9). This page will pop up after the alignment run is completed.At the top of the page, click the right button of the mouse over *Fasta format → Save link as* → *Save it adding “_final.fas” suffix* (Figure 9).

##### Trimming and Retrieving Coding Sequences (Using MEGA Version 11)

Go to the MEGA home page (https://www.megasoftware.net/, accessed on 27 January 2022), select your operating system, *Graphical GUI*, and *MEGA 11* (or newer versions if available), and click on the *Download* button.Follow the steps for MEGA downloading and installation.Open the MEGA program and load every alignment file (*ALIGN* → *Edit/Build Alignment → Open a saved alignment version → OK → Open the downloaded file*) (Figure 10).Trim the sequences using the *Scissors* tool (Figure 10) or using *Ctrl* or *Cmmd + X*. If you are using genomic or RNA data, you can click on *Translated Protein Sequences* to obtain the deduced amino acid sequences (see also Section 3.2.1), which can be useful for delimiting open reading frames (identification of start/stop codons and intronic/exonic regions by visual inspection).Finally, click on *Data → Export Alignment → Fasta Format → Save it.*When working with genomic sequences, it may be necessary to conduct steps in Section 3.1.5 several times (iterative refinement) in order to eventually obtain the coding sequence of interest. The alignment becomes progressively refined by iteratively trimming intronic regions and leftover positions at the beginning and the end of the sequence.In some cases, you will have to rerun *get_RAW_sequences.py*, changing the *--**in_len* parameter value in order to cover all the sequence. Sometimes the chosen value may be too small, and part of the sequence can be left out unintentionally.Once the coding sequence has been fully identified, save the alignment with MEGA as detailed above and add the suffix, “_final.fas”.

### 3.2. Phylogenetic Analyses

#### 3.2.1. Translate Sequences

A.Locally Using Python

This step is mandatory if you want to obtain a protein data matrixand you are working with genomic or RNA sequences.In the Spyder console, type *run translate_sequences.py [data_type]*. Typically, you are going to use it with genomic sequences or, in some cases, with RNA sequences (particularly for checking and removing ambiguously aligned positions).By default, the program will look for files with “final.fas” or “RAW.fas”. Change this using the *--pattern* parameter.If you are working with several species that each have a different genetic code, you will have to run this program several times. We recommend cutting those folders that share the same genetic code and pasting them into a new folder. Use *--directory* arguments to indicate the new path and run the program. Then, put the folders back in their initial location and repeat the step with the different genetic codes.

B.Locally Using MEGA

Open the MEGA program (see Section 3.1.5. Trimming and Retrieving Coding Sequences for MEGA installation).Import alignment as in Section 3.1.5. Trimming and Retrieving Coding Sequences (Figure 10).On the emergent window, click on *Translated Protein Sequences → Choose the adequate genetic code* (Figure 10).Finally, click on *Data → Export Alignment → Fasta Format → Save with “_translated.fas” suffix.*Repeat with all the files that need to be translated.

#### 3.2.2. Matrix Assembly

A.Locally Using Python

In the Spyder Python console, type *run get_combined_seqs.py [output name][data_type]*. The three data types can be combined. However, note that if you have information for more than one data type for the same species, then you may obtain redundant sequences.By default, the pattern for file searching is “final.fas”. However, it is programmed to look for “final_translated.fas”, “RAW.fas”, and “RAW_translated.fas” when it cannot find the first pattern.You can change this using the *--pattern* parameter. Then, the program will search for “new_pattern.fas”, “new_pattern_translated.fas”, “RAW.fas”, and “RAW_translated.fas”.The program will produce a data matrix named “[output name]_all_combined.fas”, which will be downloaded to the *Data* folder.

B.Locally Using MEGA

Import a sequence file as in Section 3.1.5. Alignment B. (Figure 10).In the emergent window, click on *Edit → Insert Sequence From File → Open every sequence file* (Figure 10).This step can also be conducted manually with a text editor; simply open every file containing the downloaded sequences and paste their content into a new file, one after the other.

#### 3.2.3. Alignment

See Section 3.1.5. Alignment. If using the Python version, use *–directory*, and drag the combined matrix file.

##### Alignment Filtering

A.Using trimAl Locally through Python

Go to http://trimal.cgenomics.org/downloads (accessed on 27 January 2022) and download the specific program according to your operating system.Decompress the trimAl file. For Windows and Mac/Unix users, open the terminal and follow the steps on the trimAl README.md file. You can run the program on the terminal following the instructions at http://trimal.cgenomics.org/use_of_the_command_line_trimal_v1.2 (accessed on 27 January 2022) or the following steps.Type *run alignment_trimming.py [path_to trimAl] [path_to_matrix] [combined_matrix]* in the Spyder console. For Windows users, trimAl will be stored in the *bin* directory. For Mac and Unix users, trimAl will be stored in the *source* directory.This will remove all positions in the alignment with gaps in 90% or more of the sequences (-gt 0.9), which is the default option of the program.The trimming algorithm can be changed using --trimal_command. See http://trimal.cgenomics.org/use_of_the_command_line_trimal_v1.2 (accessed on 27 January 2022) for additional information.

B.Using trimAl through the Phylemon Web Server

Go to http://phylemon2.bioinfo.cipf.es/, accessed on 27 January 2022.Create and account and login or star as anonymous user.Go to *Utilities* → *Alignment Utilities* → *TrimAl* (Figure 11).Paste the content of the matrix or upload the file clicking on *browse server → Upload new file → Browse → Open the matrix → Select format → Aligned sequences → Upload → Accept.*Click on *Method → User define →* In *Gap threshold, fraction of positions without gaps in a column* set 0.1. Similar output as using Python version.Run the job.Refresh the page. On the right panel, open the job when finished.

#### 3.2.4. Phylogeny Inference

A.Using IQ-TREE Locally through Python

Go to http://www.iqtree.org/#download (accessed on 27 January 2022) and download the adequate version for your operating system.Decompress the folder.Type *run phylogenetic_inference.py [iqtree_folder] [trimmed_matrix] [data_type].*IQ-TREE program is located in the *bin* folder in the IQ-TREE program folder. Drag this folder to the Spyder console when running.The “.treefile” will be stored at *Data/Phylogenetic_inference*.

B.Using IQ-TREE Web Server

Go to http://iqtree.cibiv.univie.ac.at/, accessed on 27 January 2022.Browse the trimmed matrix in the *Alignment file* field.Select sequence type.Do not change any more parameters for a similar run and output as the Python version.Enter your e-mail and click on *SUBMIT JOB.*When finished, you will receive and e-mail. Click on the provided link, and with the button on the left, click *DOWNLOAD SELECTED JOBS.*

##### Tree Visualization

Go to the FigTree website (https://github.com/rambaut/figtree/releases (accessed on 27 January 2022)) and download the specific version for your operating system.Decompress the file and open FigTree (Figure 12).Click on *File* → *Open* → *Open “.treefile” file.*Provide a label for the values of support (or leave unchanged).The left panel allows for modification of multiple tree features displayed as collapsible menus. For example, tree appearance options can be changed from the *Appearance* menu.Display the values of support. Click on *Node Labels* → *Display* → *Select the name of the label provided before* (Figure 12).The root of the tree can be changed by selecting a specific branch and then clicking the *Reroot* button at the top of the window.

## 4. Expected Results

The detailed protocol will primarily yield a sequence alignment and a phylogenetic tree (along with a best-fit model of substitution) for a particular group of organisms of interest. Altogether, these resulting data are the typical starting point of molecular evolution analyses, such as those aimed at detection of adaptive and/or purifying selection, ancestral protein reconstruction, or protein structure. Additionally, the protocol can provide refined annotation of raw genome sequences, which can be used for other goals, such as gene mapping or functional analyses. Finally, the provided pipeline and protocol constitute a basis for bioinformatic work with DNA and protein sequences that can be easily modified and adapted for many other specific tasks in the present context of massive generation of genomic data.

## 5. Troubleshooting

The first issue that readers may encounter relates to software versions and maintenance of websites and online resources. Even if some of these are discontinued, the methods outlined in the workflow would stand as a solid guideline. Every day, new applications and versions are released, and it is not difficult to find programs that perform similar tasks to the ones mentioned in the protocol. It is also possible that the functions of some of our Python applications may become deprecated in the future. This may represent a more challenging issue because the user might have to modify the source code (therefore requiring at least some basic notions of programming languages). The user should also be aware that unexpected inputs and commands can result in illogical outcomes. It is therefore of critical importance to follow the instructions provided here or in the README.md file and to carefully inspect the output files generated in each step.

Additionally, there may be internet connection issues when downloading a large number of genomes (especially genomic sequences) using GNFish that may produce partly empty or incomplete files (apparently correct sometimes). This will prevent further progression in the workflow, as subsequent steps become logically impossible even if the inputted commands are valid. A similar issue can occur when downloading the query sequences, although this is less common.

Generally speaking, it is always a challenge to work with genomic sequences because their analysis usually requires substantial investment of time and use of computational resources. If possible, we recommend running the GNFish package on an HPC (high-performance computing) cluster, particularly when working with scaffold sequences. As previously stated, these data must be iteratively curated, and identifying open reading frames can be arduous or even impractical in some difficult cases. In all cases, outcome sequences must be carefully examined by the user. The more information about the gene is available, the easier and more accurate the retrieval becomes. In some particular cases, the user would benefit from repeating the retrieval process if additional knowledge of the data is gained.

Another issue may be related to phylogenetic inference if the maximum likelihood analysis becomes trapped in local optima (particularly an issue for very large datasets). If this is suspected, the IQ-TREE developers recommend repeating the analysis with at least 10 independent runs. 

Last but not least, high-throughput sequencing technologies are not flawless, and sequencing errors may occur in the source data. We recommend using the RefSeq database throughout the protocol because it only contains curated sequence data. We also suggest the use of some filters when searching for sequences, which may improve the quality of the outcome. Nonetheless, the user should be aware that full exclusion of low-quality sequences may not always be possible, and dealing with a certain proportion of them in the datasets is likely unavoidable. In principle, this proportion should be low, with little or a negligible impact on subsequent analyses.

## Figures and Tables

**Figure 1 mps-05-00026-f001:**
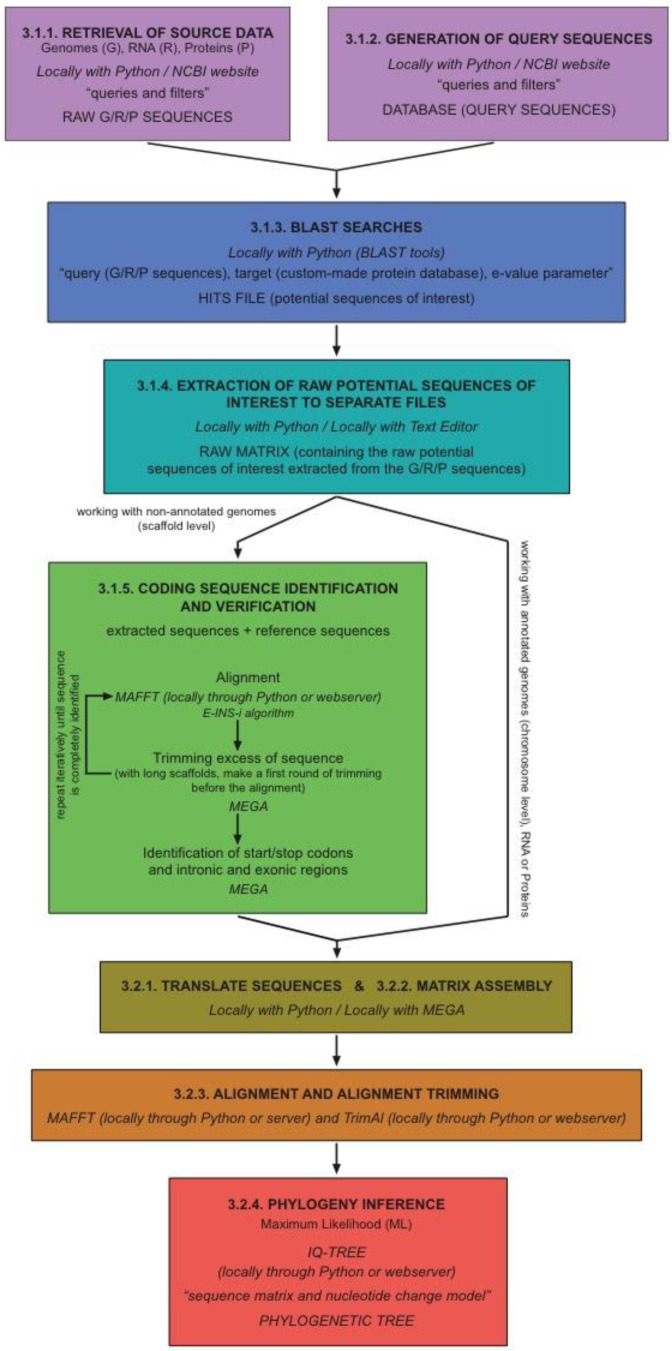
Schematic flowchart of the main steps of the protocol.

**Figure 2 mps-05-00026-f002:**
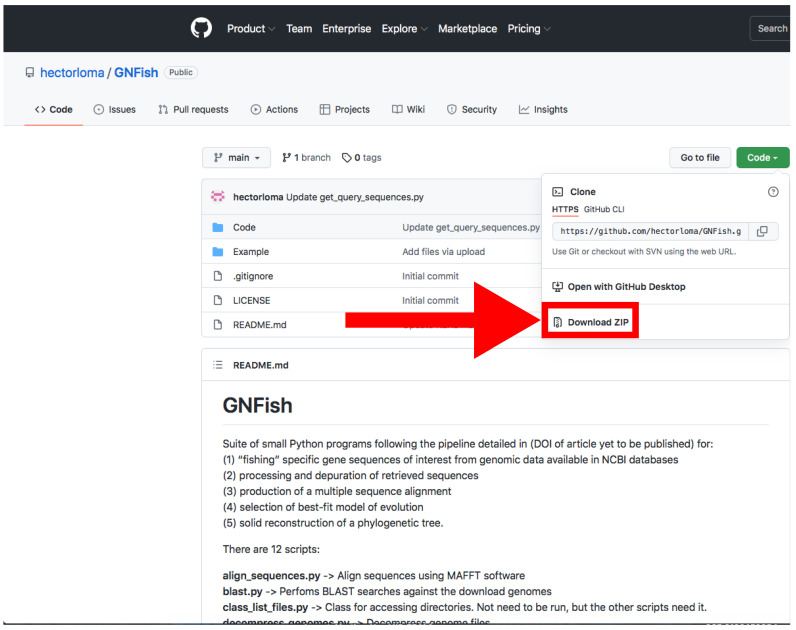
GitHub web server where the GNFish package is stored. The red arrow indicates the button for downloading. The Code button appears in green (top right). Accessed on 27 January 2022.

**Figure 3 mps-05-00026-f003:**
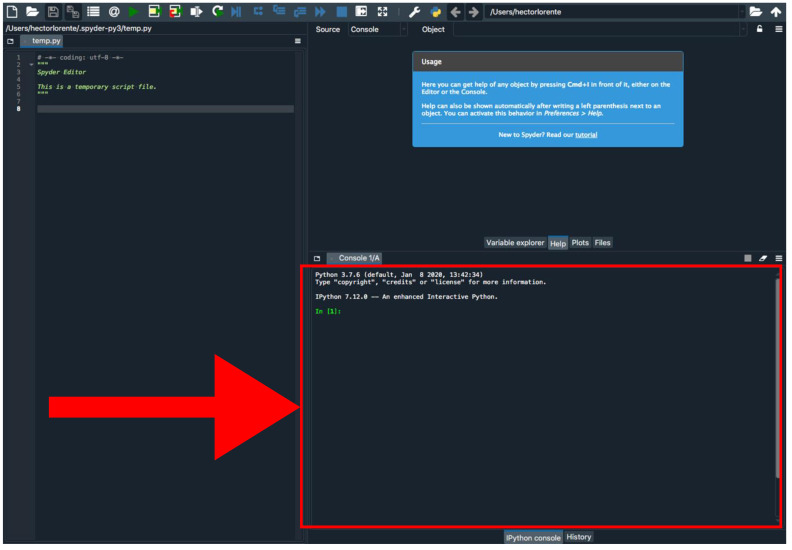
Spyder program interface. The red arrow indicates the Python console.

**Figure 4 mps-05-00026-f004:**
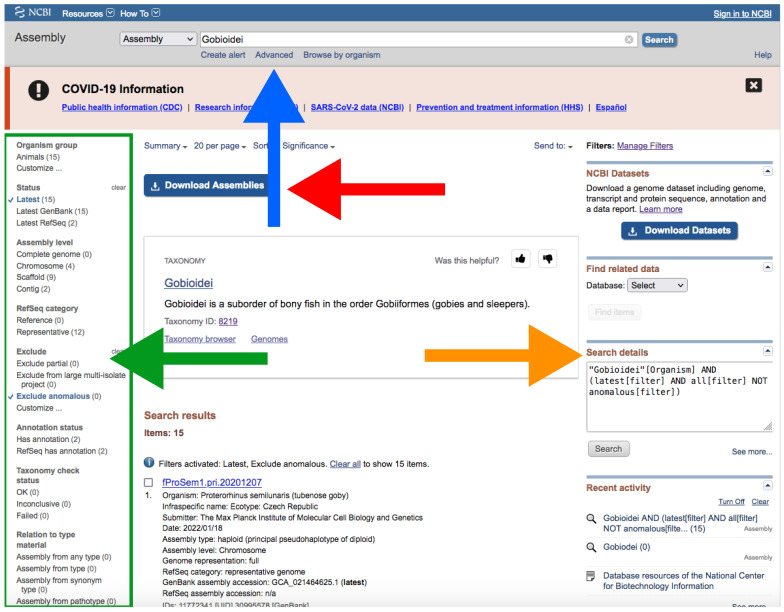
NCBI web server interface after conducting a search on assembly database. The blue arrow indicates the search box; the green arrow and green square indicate *Representative* and the filters side panel, respectively; the orange arrow indicates “Search details”; the red arrow indicates the *Download Assemblies* button (see Figure 5 for more information about downloading). Accessed on 27 January 2022.

**Figure 5 mps-05-00026-f005:**
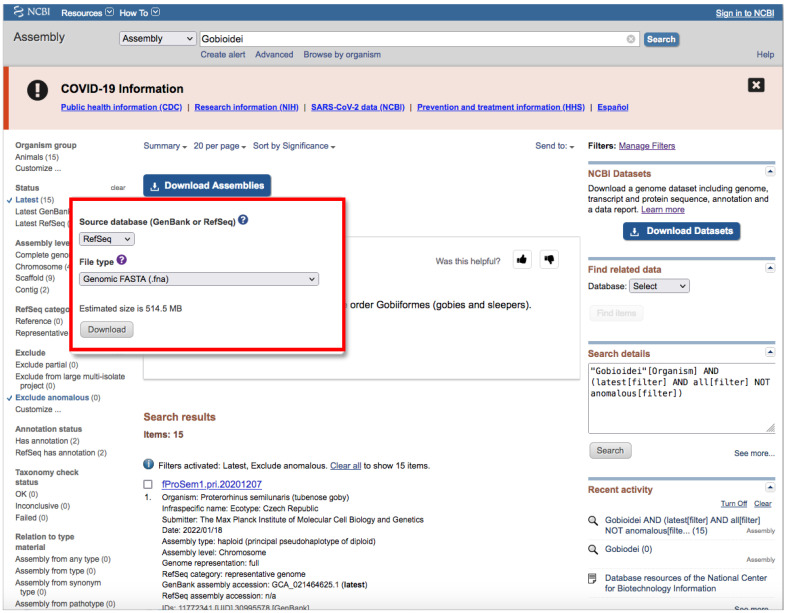
*Download Assemblies* button in detail. Accessed on 27 January 2022.

**Figure 6 mps-05-00026-f006:**
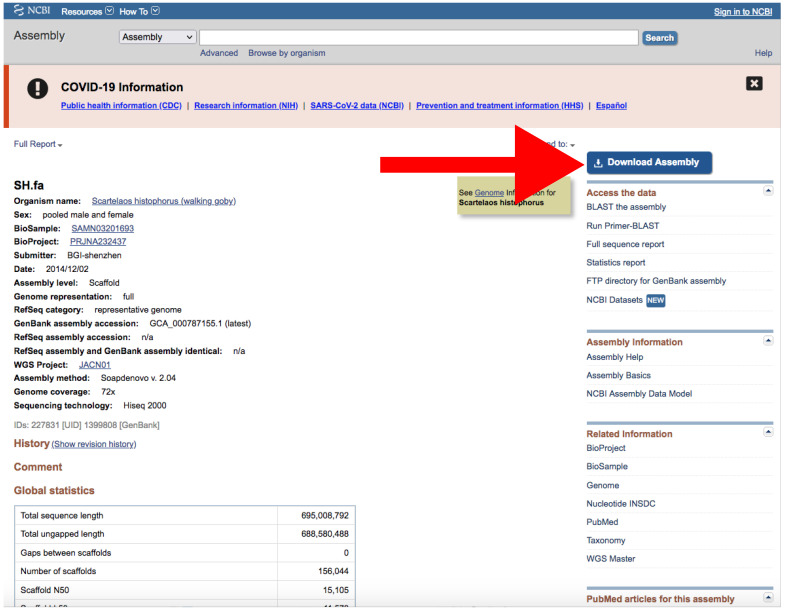
Example of an assembly entry. The NCBI web server will automatically redirect to a page like this if there is only one assembly that matches your query parameters. The red arrow indicates the *Download Assembly* button. Accessed on 27 January 2022.

**Figure 7 mps-05-00026-f007:**
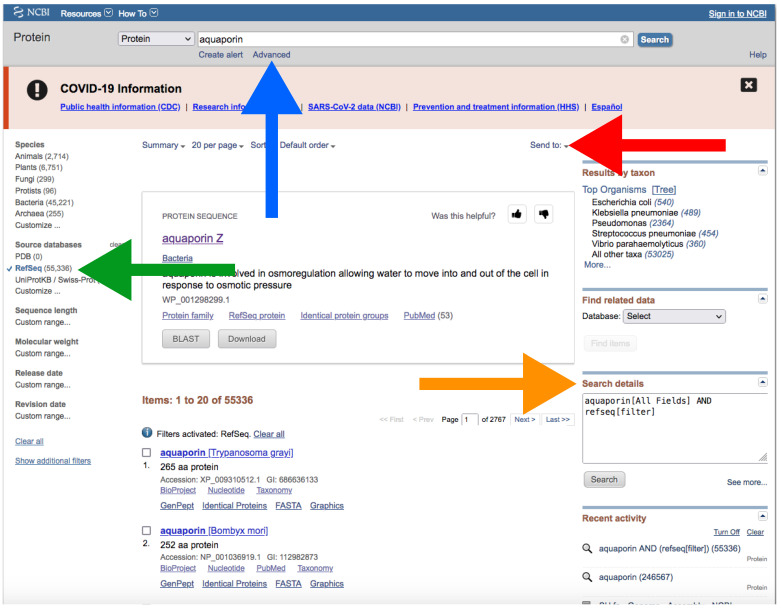
The NCBI web server interface after search in the protein database. The blue arrow indicates the search box; the green arrow and green square indicate “RefSeq” and the filters side bar, respectively; the orange arrow indicates “Search details”, and the red arrow button and red square indicate the “Send to” button and the “Create file” window for downloading, respectively. Accessed on 27 January 2022.

**Figure 8 mps-05-00026-f008:**
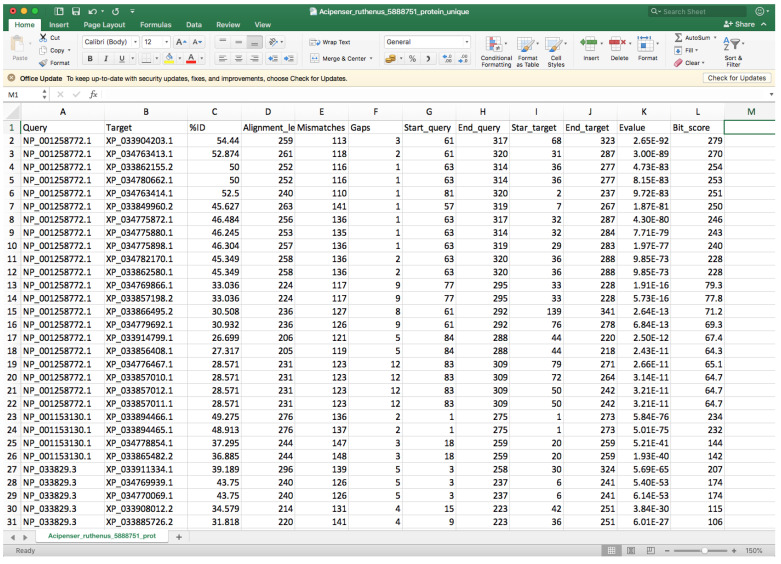
Example of a BLAST output file viewed in Excel.

**Figure 9 mps-05-00026-f009:**
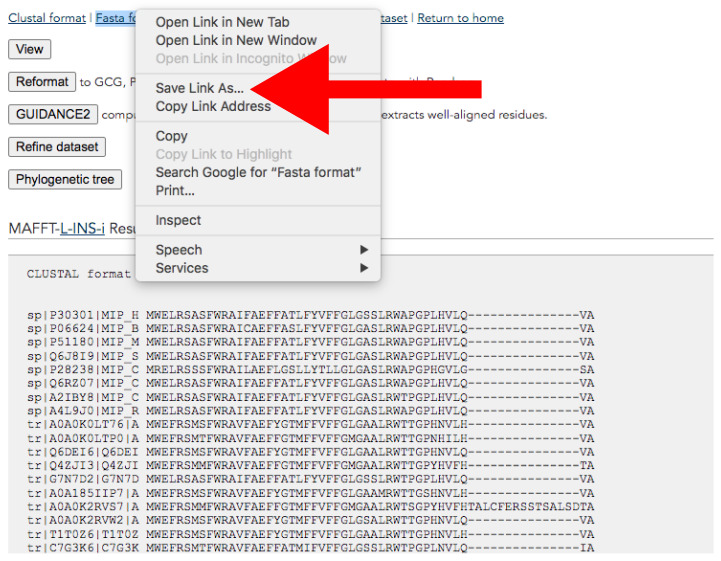
MAFFT result web page. The red arrow indicates *Save link As…* button for downloading. Accessed on 27 January 2022.

**Figure 10 mps-05-00026-f010:**
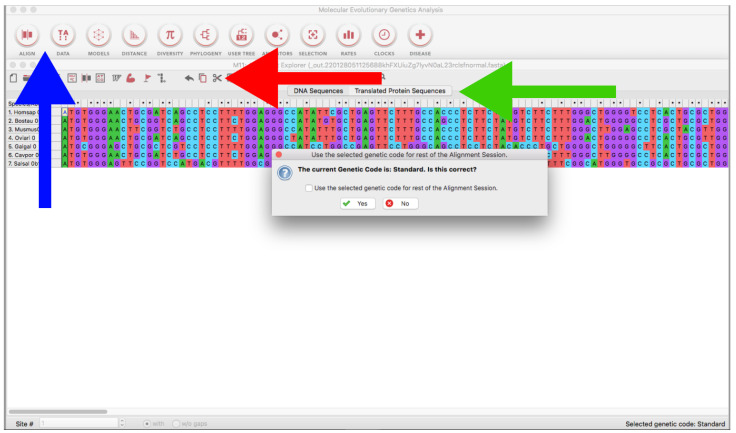
MEGA version 11 program showing a nucleotide alignment window. The blue arrow indicates the tool bar where the *ALIGN* and *DATA* buttons are located. The red arrow indicates the scissors tool. The green arrow indicates the *Translated sequences* tab.

**Figure 11 mps-05-00026-f011:**
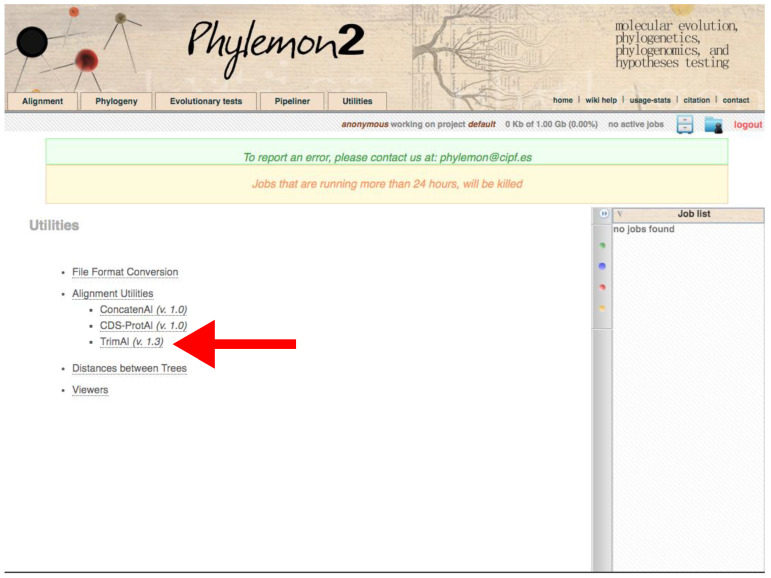
Phylemon2 *Utilities* web server. The red arrow indicates the trimAl program. Accessed on 27 January 2022.

**Figure 12 mps-05-00026-f012:**
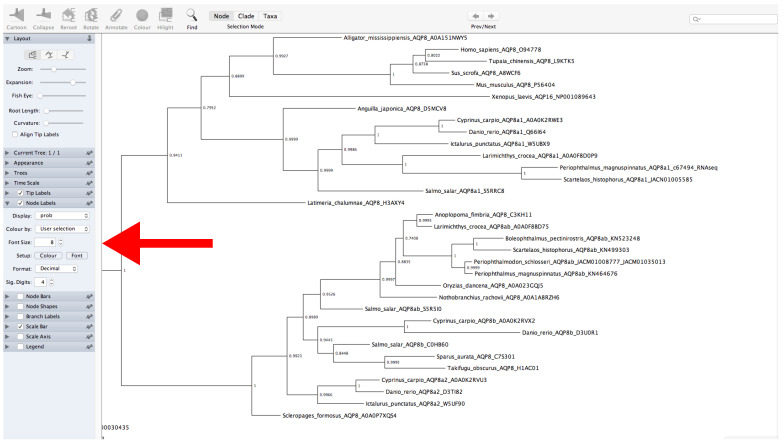
FigTree program. The red arrow indicates the *Node Label* panel.

## Data Availability

The GNFish package containing code scripts, readme file, and examples is available from the GitHub platform at https://github.com/hectorloma/GNFish, accessed on 27 January 2022.

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
