# Peer review of "Genomic Fishing and Data Processing for Molecular Evolution Research"

_mps, 2022, doi:10.3390/mps5020026_

Round 1

Reviewer 1 Report

Methods and Protocols is a journal to describe experimental techniques in biomedical sciences. In this manuscript, Mauro and co-workers described a workflow to prepare genome-extracted datasets for molecular evolution research. I appreciate the level of detail they provide in the MS and believe this article should be a good guideline for relevant research. Below are some comments to further improve the quality/readability of this MS. I suggest a minor revision for this manuscript.

  1. As a protocol, the current MS is missing the troubleshooting section. I would suggest the authors point out a few common errors the readers may make when adapting the workflow – this will greatly improve the readability/reproducibility of this protocol.
  2. The introduction section is a little too brief and lacks sufficient citations of relevant papers. Please add more introductory paragraphs/citations to highlight the usefulness of described protocol, for example, if a similar protocol has been used in any of the recent studies in the field of molecular evolution research.
  3. In the introduction, the authors mentioned that ‘this protocol is intended to be operated with command-line interfaces on Linux/Unix’. However, the screenshots (e.g. Fig. 3 and Fig. 8) indicate they were using Mac. Please revise the introduction accordingly or upload screenshots under Unix.

Reviewer 2 Report

The study authored by Hector Lorente-Martinez et al. described the usage of GNFish package on sequence processing and molecular evolutionary tree construction analysis.
Overall, the manuscript was written nicely (Introduction part) and clearly explained all the procedures. The screenshots of figures showed a clear layout for conducting the study autonomously. Since, the article needs several corrections and clarifications for the improvement.

Major comments
1. The procedures are explained very simple, but the same procedures are repeatedly for the sequence process. For Ex. Why the authors were performed sequence alignment trimming using MEGA (3.1.5.1) and doing the same procedures using TrimAI server (3.2.2)?
2. Like 1st question, line 485-491 and 521-525 repeatedly informing the process of MEGA program and alignment process.
3. Abstract, line 16: Why the authors have stated "Fishing"? Is it searching sequences or mining sequences? 
4. 2.2. Phylogenetic analysis, line 221: Why the authors have used to visualize the phylogenetic tree on FigTree, because, the sequence alignment/triming process was conducted by MEGA. Why not prefer to compute and visualize using MEGA tool?
5. Line 224: Why the dentroscope tool cited, which is not applied in the procedures?
6. The authors have mentioned several times "README.md" in the article (line: 325, 330, 372, 392, 453, 530, 541, 551, 555, 580), else write a sentence about "README.md" in the 3.1.1 section, which is enough for the understanding all the process. Remove the repeated information from through out the article.

Minor comments:

1. Introduction, Line 31: "marriage of programming" change to fusion/combination/association?
2. Figure 1: title not corresponding to the main text titles "3.1.2. Generation of query sequences" but in text mentioned "Download query sequences". Similar, 3.2.1 is an error in the flowchart. It might be 3.2.2, and it is not corresponding to the main text numerical notation for the headings.
3. Line 206: Rewrite sentence " just like BLAST do" incomplete.
4. Line 287 and 312: There is no subheading like "3.1.1.1", correct it.
5. A lot of numeric subheadings are making confusion to the readers. Line 506-507: There is an error on "3.2.6.1" section numeric notation. There is no a subheading on 3.2.5 exist in the article.
5. Line 547: Error on subheading numbers "3.2.6.3", but I assume that is "3.2.2. for alignment" and 3.2.3 for Phylogenetic inference as corresponding to the Figure.1 flowchart.

Reviewer 3 Report

The authors have proposed a comprehensive workflow to prepare the datasets derived from the “multi-omics” studies to fulfill the fundamental requirements of the molecular evolution analyses. The authors have provided sufficient background as well as detailed explanations of each step in the workflow with some sample software suggested. I think this is a very interesting paper. I believe this paper would make significant contribution to the related areas by helping the users format specific workflow in their research activities. I have no major technical concerns but a few editorial suggestions:

Lines 227 and 228 showing the same subsection number (3.1.1.). Overall, the numbers of the many subsections of section 3.1. need to be re-organized.

I have seen that some of the figures are screen shots from the internet. I wonder if it is necessary to indicate approximately the dates when these websites were accessed by the authors because this may help the users realize that the display may not look exactly the same as shown in this manuscript.

Figure 10: the version of MEGA needs to be indicated, again, because the displays of different versions of MEGA look different.

Round 2

Reviewer 2 Report

Really, good appreciation to the authors. They have made all the corrections and changes. Provided clear responses.

I am satisfied with the revision.